# INVESTIGATING FACTUALITY IN LONG-FORM TEXT GENERATION: THE ROLES OF SELF-KNOWN AND SELF-UNKNOWN

## ABSTRACT

Large language models (LLMs) have demonstrated strong capabilities in text understanding and generation. However, they often lack factuality, producing a mixture of true and false information, especially in long-form generation. In this work, we investigates the factuality of long-form text generation across various large language models (LLMs), including GPT-4, Gemini-1.5-Pro, Claude-3-Opus, Llama-3-70B, and Mistral. Our analysis reveals that factuality scores tend to decline in later sentences of the generated text, accompanied by a rise in the number of unsupported claims. Furthermore, we explore the effectiveness of different evaluation settings to assess whether LLMs can accurately judge the correctness of their own outputs: Self-Known (the percentage of supported atomic claims, decomposed from LLM outputs, that the corresponding LLMs judge as correct) and Self-Unknown (the percentage of unsupported atomic claims that the corresponding LLMs judge as incorrect). The results indicate that even advanced models like GPT-4 and Gemini-1.5-Pro fail to achieve perfect Self-Known scores, while their Self-Unknown scores remain notably above zero, reflecting ongoing uncertainty in their self-assessments. Moreover, we find a correlation between higher Self-Known scores and improved factuality, while higher Self-Unknown scores are associated with lower factuality. Interestingly, even without significant changes in the models' self-judgment (Self-Known and Self-Unknown), the number of unsupported claims can increases, likely as an artifact of long-form generation. These findings show the limitations of current LLMs in long-form generation, and provide valuable insights for improving factuality in long-form text generation.

## 1 INTRODUCTION

The long-context capabilities of large language models (LLMs) (OpenAI, 2023b; AI@Meta, 2024; Jiang et al., 2024; GeminiTeam, 2024; Anthropic, 2024) have seen significant advancements in recent years. Lots of work (Shaham et al., 2023; Bai et al., 2024; An et al., 2024; Zhang et al., 2024; Kuratov et al., 2024) have explored the ability of LLMs to handle long contexts, however, relatively few have examined their ability for long-form text generation.

Despite LLMs have the impressive generative abilities, these models are prone to producing hallucinations (Li et al., 2023; Min et al., 2023) where the generated content often blends factual and fabricated information. This tendency not only undermines performance but also poses substantial risks in practical applications. To assess the factuality of responses from LLMs, recent research (Fan et al., 2020; Wright et al., 2022; Min et al., 2023; Manakul et al., 2023) has introduced a method that breaks down generations into atomic claims – short statements each containing a single piece of information. These atomic claims are then individually evaluated to determine whether they are supported by evidence or unsupported.

To ensure the reliable use of LLMs, it is also crucial that they possess the ability to recognize not only "what they know" but also "what they don't know." Recent studies, such as those by Kadavath et al. (2022); Liu et al. (2022); Guerreiro et al. (2023), have shown that language models can assess the validity of their own claims. However, Srivastava et (2023); Yin et al. (2023) have pointed out the limitations of LLMs in recognizing their own knowledge gaps.

In this work, we investigate the factuality patterns of long-form text generation across various LLMs. We first assess the factuality of long-form generation at different relative positions using two annotated datasets and two models: ChatGPT and PerplexityAI (which integrates a search engine). Our findings indicate that sentences generated earlier in the sequence generally demonstrate higher factuality. However, these later-generated sentences also contain more unsupported claims and fewer supported claims.

We further examine the effectiveness of different evaluation settings to assess whether LLMs can accurately evaluate the correctness of atomic claims in their own generated outputs. To quantify the corresponding models' ability to judge the correctness of atomic claims, we calculate two metrics: the **Self-Known score** (the percentage of supported atomic claims judged as correct by the LLMs) and the **Self-Unknown score** (the percentage of unsupported atomic claims judged as incorrect by the LLMs). Our exploration includes three methods, notably a novel approach where the final answer option is replaced with "None of the above". This modification appears to provide a more accurate measure of the LLMs' abilities, as evidenced by a higher flip rate for supported claims and an increasing flip rate at later relative positions. This suggests that the model reassesses its confidence when faced with an option signaling uncertainty. In contrast, the low flip rate for unsupported claims indicates a consistent judgment of their incorrectness. These results suggest a nuanced understanding by LLMs of supported versus unsupported claims and underscore the importance of specific evaluation settings to accurately gauge model performance. Our findings align with human annotations for two LLMs, although some discrepancies, particularly with the PerplexityAI model, suggest gaps in estimation.

The main contributions of our work are as follows:

1. We explored the factuality patterns of long-form text generation across various model families (GPT-4, Gemini-1.5-Pro, Claude-3-Opus, Llama-3-70B, and Mistral). We found that even the most advanced LLMs typically exhibit lower factuality scores in the later segments of long-form text. Retrieval-Augmented Generation (RAG) systems show a similar trend, although they tend to maintain higher factuality overall.

2. We analyzed Self-Known and Self-Unknown ratios for these LLMs across different segments of their own generated texts. The results showed relatively higher Self-Known scores; however, even strong LLMs (GPT-4, Gemini-1.5-Pro, Claude-3-Opus, etc.) generally achieved only about 50% on the Self-Known score. The Self-Unknown scores were significantly above zero. These findings indicate that even the most advanced LLMs still produce outputs with limited self-acknowledgment ability.

3. We developed a mathematical framework linking Self-Known and Self-Unknown scores to factuality, providing deeper insights into their relationship. Both empirical and theoretical results demonstrate that higher Self-Known scores correspond to improved factuality, while higher Self-Unknown scores are associated with reduced factuality. Moreover, even without significant changes in the models' self-judgment (Self-Known and Self-Unknown), the number of unsupported claims may still increase, likely reflecting the inherent challenges of long-form generation.

## 2 LONG-FORM TEXT GENERATION

To evaluate the factuality of LLM responses, recent work (Liu et al., 2023; Chen et al., 2022; Min et al., 2023) breaks a generation into a series of atomic claims—short statements that each contain one piece of information. Each atomic claim is then individually evaluated to determine whether it is supported or unsupported. In this section, we first explore the factuality patterns of these atomic claims in long-form text generation.

### 2.1 OBSERVATIONS

In order to explore the factuality of long-form generation at different relative positions, we use the human annotated data from Min et al. (2023) to compute the macro-average percentage of three different claims (supported, unsupported, and irrelevant) across five different relative positions. In their human-annotated data, each long LLM generation is decomposed into atomic claims and each atomic claim is assigned with one of the three labels ("supported", "not-supported", "irrelevant").

The detailed procedures for computing fractions of different type claims at different relative positions are as following:

1) For every sentence in a generation, we computed the fraction of the number of supported atomic claims, unsupported atomic claims and irrelevant claims; 2) We got the relative position of each sentence, e.g., if it is the third sentence out of six, its relative position would be 3/6 = 50%; 3) We then grouped all sentences that fall within the same relative position range: 0-20%, 20%-40%, 40%-60%, 60%-80% and 80%-100%; 4) Finally, we computed the macro-average percentage within the same relative position group.

Figure 1 (a) shows the ChatGPT results (Figure 8 in the Appendix shows the PerplexityAI results.). We can see that unsupported claims percentage is higher when these sentences are generated later. We hypotheses the possible reasons are the error propagation and these generated claims are with low confidence by LLMs.

We also compute the number of different type claims at different relative positions with the above similar procedures. As shown in Figure 1 (b), LLMs tend to generate less reliable information as they continue the generation. More unsupported claims are included in the continued generation.

**Open Questions.** Do LLMs recognize that supported claims are indeed supported, and that unsupported claims are unsupported? Do LLMs better identify unsupported claims that appear later in the text compared to those that appear earlier?

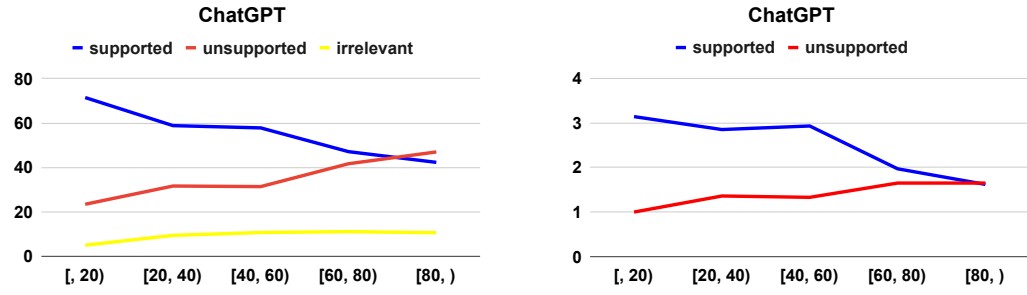

(a) Percentage (%) of supported, unsupported and irrelevant atomic claims.

(b) Number of supported and unsupported atomic claims.

Figure 1: Long-form generation across different relative positions (%) for ChatGPT.

## 3 SELF-KNOWN AND SELF-UNKNOWN

To investigate these questions, we examine whether the corresponding LLMs recognize their atomic claims by computing two metrics: **Self-Known** (the percentage of supported atomic claims that the corresponding LLMs judge as correct) and **Self-Unknown** (the percentage of unsupported atomic claims that the corresponding LLMs judge as incorrect). While there is related work, such as Rajpurkar et al. (2018); Xiong et al. (2024), our approach differs in two key ways: (1) Evaluation is conducted on atomic claims, which are derived from sentences in long-form generation, rather than assigning a score to the entire model output; (2) Our focus is on factuality (whether an atomic claim is true or false), rather than on uncertainty scores (i.e., "How likely is the above answer to be correct?").

We explore the computation of **Self-Known** and **Self-Unknown** using the following three approaches ( with the corresponding prompt templates provided in Appendix Section A.2):

- **Direct-Asking**: In this approach (Rajpurkar et al., 2018), the atomic claim is directly given to the corresponding LLMs and be asked whether the statement is true or false.
- **Question-Answering**: Given an atomic claim, a question-answer pair can be derived (Trischler et al., 2017; Rajpurkar et al., 2018; Hu et al., 2024) with GPT-4 Turbo. For example, "Lanny Flaherty is an American." can be used to derived a question-answer

pair ("What nationality is Lanny Flaherty?", "American"). Then, given the question and answer, we ask the corresponding LLMs whether the answer is true or false.

- **Question-Answering w/NOA**: Similar to the above approach, a question-answer pair is derived according to each atomic claim. One big different is: given question and answer, one more addition choice ( "None of the above") (Rajpurkar et al., 2018) is given to the corresponding LLMs. This is a well-defined evaluation because it can check whether the model actually knows the answer of the question, especially if the question is vague or context-information is missing.

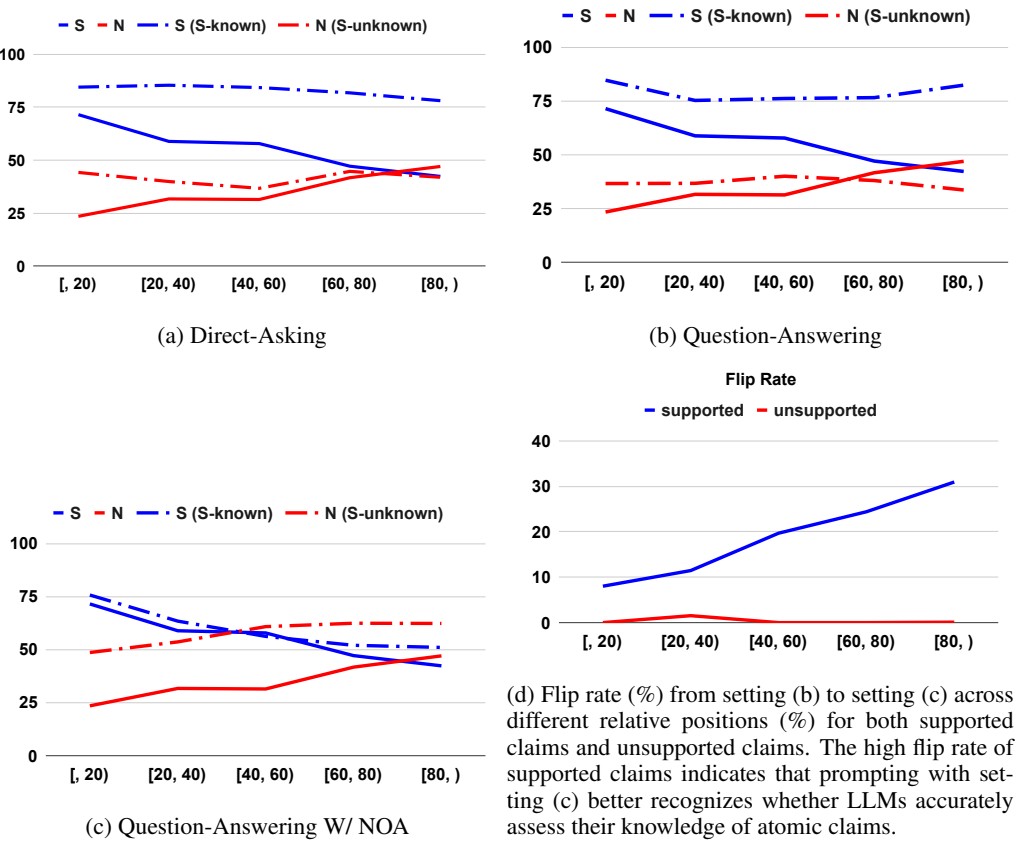

(a) Direct-Asking

(b) Question-Answering

(c) Question-Answering W/ NOA

(d) Flip rate (%) from setting (b) to setting (c) across different relative positions (%) for both supported claims and unsupported claims. The high flip rate of supported claims indicates that prompting with setting (c) better recognizes whether LLMs accurately assess their knowledge of atomic claims.

Figure 2: Self-Know and Self-Unknown results of ChatGPT across different relative positions (%). **S**: **factuality** (percentage of supported atomic claims); **N**: percentage of unsupported atomic claims; **S (S-known)**: **Self-Known** score; **N (S-unknown)**: **Self-Unknown** score

We compute the Self-Known score and the Self-Unknown score using these prompt templates. The human annotated data on ChatGPT[1] are used in this experiments. Figure 2 presents the results on ChatGPT.

**Comparison on the above three evaluation settings** With the first two settings, the results of **Self-Known** score and **Self-Unknown** score are similar. However, the results of the third setting differ from the other two. We hypothesize that the reason is that the added choice, "None of the above" which allows the LLM to determine whether it knows the answer to the question.

To examine the effect of this setting, we plot the flip rate (claims judged as correct by the LLM in setting (b) but judged as incorrect in setting (c)) for supported and unsupported claims. As shown in Figure 2d, there is a high flip rate for supported claims, and this rate increases with higher relative positions. In contrast, there is almost no flipping for unsupported claims. Therefore, setting (c) is

---

[1]The labeled ChatGPT data is also from Min et al. (2023) as above. There are 183 long generations of ChatGPT.

more suitable for checking whether the LLM knows a atomic claim. The high flip rate observed for supported claims suggests that the model is reconsidering its initial judgments when presented with the option "None of the above". This indicates that the model may not be entirely confident in its original answers and is more likely to recognize uncertainty. The increasing flip rate for higher relative positions further supports this, implying that the model's confidence decreases as the position of the claim within the context changes.

In summary, we observed similar results between setting (a) (Direct-Asking) and setting (b) (Question-Answering), and a significant difference between setting (b) (Question-Answering) and setting c (Question-Answering W/ NOA). **The deeper analysis between setting b and setting c revealed that setting (c) recognizes atomic claims more confidently and treats atomic claims that flip as unknown. This is why we chose to use setting (c) in the subsequent experiments.**

## 4 ANALYSIS

We denote the prompt input of LLMs as $x$ and long output of LLMs as $y$. The binary auxiliary label $d = 1$ indicates the LLM output is correct and $d = 0$ indicates LLM output is wrong.

We assume that $P(d = 1 \mid y, x)$ is equal to **factuality score**[2] $\sigma$ of LLM output $y$. Given $x$, the joint distribution of between the auxiliary label and model output $(d, y)$ is

$$\sigma * P(y \mid x) \tag{1}$$
$$= P(d = 1 \mid y, x) * P(y \mid x) = P(d = 1, y \mid x)$$
$$= P(d = 1, y_{\text{correct}} \mid x)\sigma + P(d = 1, y_{\text{wrong}} \mid x)(1 - \sigma)$$
$$= P(d = 1 \mid y_{\text{correct}})P(y_{\text{correct}} \mid x)\sigma + P(d = 1 \mid y_{\text{wrong}})P(y_{\text{wrong}} \mid x)(1 - \sigma) \tag{2}$$

$y_{\text{correct}}$ refers to model outputs aligned with the ground truth and $y_{\text{wrong}}$ refers to outputs that are wrong. Because $y$ is the generated output according to the log-likelihood, the correct part and incorrect part have similar log-likelihood. Then, it is reasonable to have this following assumption:

$$P(y \mid x) \approx P(y_{\text{correct}} \mid x) \approx P(y_{\text{wrong}} \mid x)$$

Then, after cancel the above three terms in Equation 1 and Equation 2 ,

$$\sigma = P(d = 1 \mid y_{\text{correct}})\sigma + P(d = 1 \mid y_{\text{wrong}})(1 - \sigma)$$

We denote $P(d = 1 \mid y_{\text{correct}})$ and $P(d = 0 \mid y_{\text{wrong}})$ as **Self-Known** score (percentage of supported atomic claims judged as correct by LLMs) and **Self-Unknown** score (percentage of unsupported atomic claims judged as incorrect by LLMs) respectively. Once the above formula is solved, we can determine the relationship among the **factuality** score, **Self-Known** score, and **Self-Unknown** score:

$$\sigma = \frac{1 - \text{Self-Unknown}}{2 - \text{Self-Unknown} - \text{Self-Known}} \tag{3}$$

Where $\sigma$ is the factuality score.

**Factuality Vs. Self-Known Vs. Self-Unknown**   Given Self-Unknown $\in [0, 1]$ and Self-Known $\in [0, 1]$, the factuality score increases when the **Self-Known** score is increased or the **Self-Unknown** score is decreased. This matches our observations in Section3 and Figure 2 (c).

**Estimation of factuality Score**   In Equation 3, we present a method for estimating the factuality score. We use the Self-Known and Self-Unknown results of the corresponding model (ChatGPT) with configuration (c) to estimate the factuality score across different relative positions. As shown in Figure 3, our estimation closely matches the human-annotation results[3].

---

[2]This is an assumption we are making: that there is no overconfidence, and the confidence score is approximately equal to the factuality score.

[3]Due to scarcity of human-annotated, long-form LLM generation datasets, we did not show

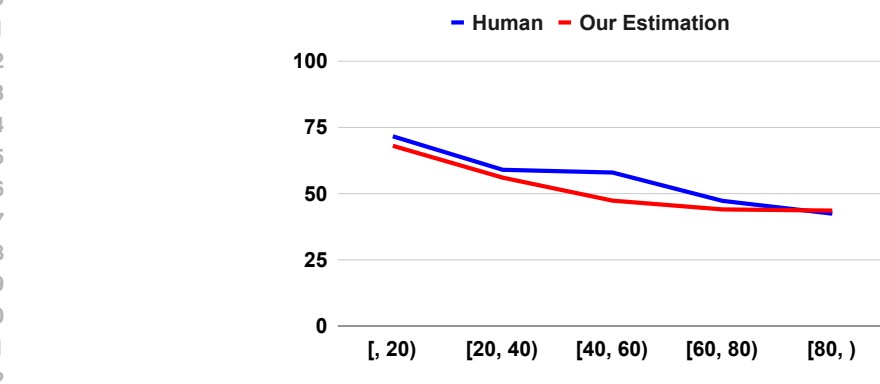

Figure 3: Human-annotation factuality score (%) and our estimation with Equation 3 across different relative positions (%).

## 5 AUTOMATIC RESULTS ON ADDITIONAL LLMS

In this section, we examine the trends in factuality, Self-Known scores, and Self-Unknown scores of other advanced LLMs using an automated evaluation tool.

### 5.1 AUTOMATIC TOOL SETTING

In Section 2, we used the human annotated data (atomic claims are short statements that are decomposed from the model's generation, and each atomic claim is labeled as either supported or unsupported based on its factual correctness.).

**Configuration** We use the tool FActScore (Min et al., 2023) for factuality evaluation with the following configuration: the latest version of GPT-3.5 (gpt-3.5-turbo-0125) is used to break a generated text into a series of atomic claims and evaluate each atomic claim against a retrieved knowledge (model name "retrieval+llama+npm" is used during the evaluation)[4].

**Results** Figure 7 in the Appendix shows the comparison between the tool's evaluation and human annotation results. We notice the tool's estimation is highly correlate well with human annotations. For number of atomic claims, the absolute difference is not bigger than 1. And the trend of tool's estimation is almost the same as human annotation. For factuality estimation, the tool's results are well-aligned with human annotations for two OpenAI models. Although there is an estimation gap for the PerplexityAI model, the trend of the estimation remains consistent with human annotations.

**Takeaway.** The tool with above configurations can well capture the trend of number of atomic claim and factuality.

### 5.2 ADDITIONAL LLMS

In this section, we explore the factuality of long-form text generation across different relative positions using automatic tools.

#### 5.2.1 EXPERIMENTAL SETUP

For each LLM, we follow four key steps to obtain experimental results: (1) generating text outputs; (2) filtering the generated content; (3): evaluating factuality; and (4): estimating **Self-Known** and **Self-Unknown** scores with the corresponding LLM. For more details on each step, please refer to Section A.5.

---

[4]In the original work, text-davinci-003 was used to get atomic claims and ChatGPT is used to evaluate whether each atomic is supported or unsupported.

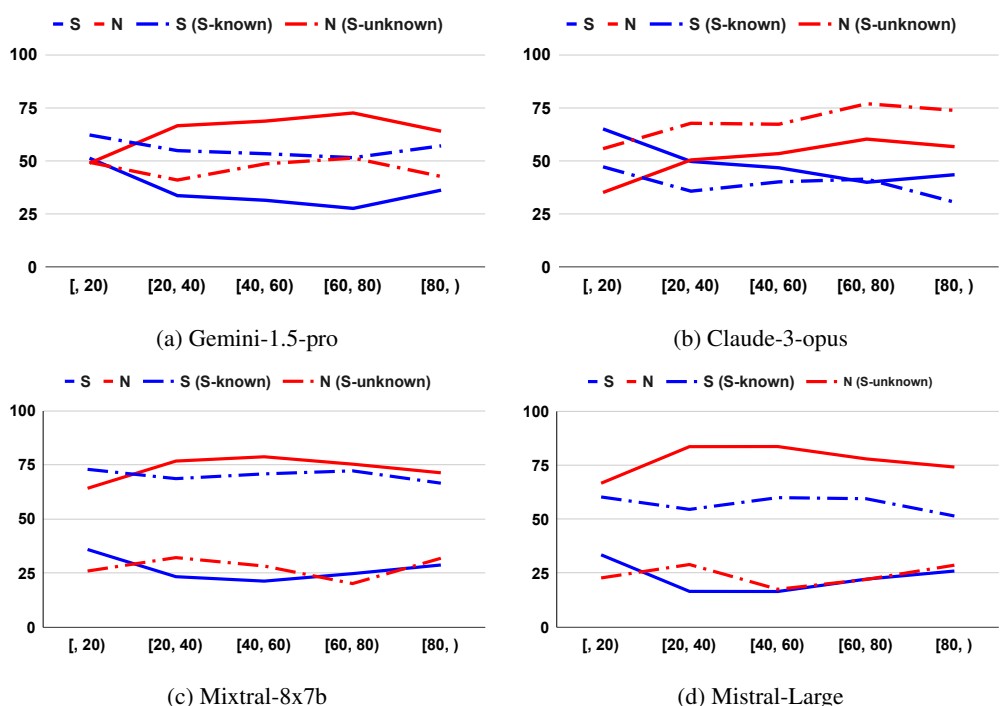

Figure 4: Self-Know and Self-Unknown results of different LLMs across different relative positions (%). **S**: **factuality** (percentage of supported atomic claims); **N**: percentage of unsupported atomic claims; **S (S-known)**: **Self-Known** score; **N (S-unknown)**: **Self-Unknown** score.

### 5.2.2 RESULTS

Table 1 presents two results for various LLMs: the average number of atomic claims per generation and the filtered rate. The filtered rate represents the percentage of instances where the LLMs do not provide valuable responses, often due to perceiving insufficient information to generate a meaningful answer. We notice that the behavior of Claude-3-opus and Gemini-1.5-pro is more conservative. These models frequently decide not to provide a valuable response, instead stating something like "I do not have enough verified information". Figure 4 show results of several powerful LLMs (GPT-4, Gemini-1.5-pro, Claude-3-opus, Llama-3-70B-Instruct, and two Mistral AI models)[5].

**Decreasing Factuality: Strong Start, Later Decline**  According to the bold blue lines in Figure 4, we observe the highest factuality scores are observed at the beginning of the generated text across all relative positions.

**Factuality Vs. Self-Known Vs. Self-Unknown**  Overall, we observe that the Self-Known score is positively correlated with factuality, as indicated by the two blue lines, and the Self-Unknown score is positively correlated with the percentage of unsupported atomic claims, as shown by the two red lines in each figure. For these advanced LLMs, the trend of these three scores across different positions shows smaller variation.

**Clear Difference in the Number of Unsupported Claims Across Positions**  In Figure 4 (e) and (f), observed minimal differences in factuality for the two models (Mixtral-8x7b and Mistral-Large). However, as depicted in Figure 5, the number of unsupported claims increases significantly from the beginning to the end of the generated text. It indicates the challenges of long-form generation. This also highlights a limitation in relying solely on factuality scores for evaluation.

---

[5]Additional LLM results are provided in Figure 9 in the Appendix.

**No Significant Changes in Self-Judgment for Some Advanced LLMs**  We can observe that there is no big change according to dashed lines (Self-Known and Self-Unknown) in Figure 4. However, the number of unsupported claims are increasing as shown in Figure 5.

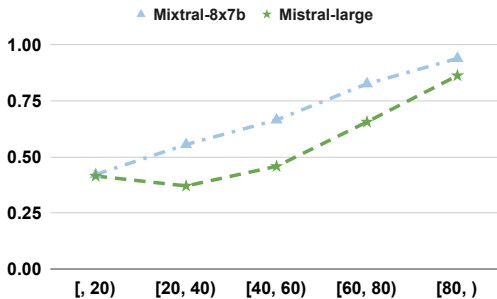

Figure 5: There may be minimal change in the factuality score, but a significant increase in the number of unsupported claims across different relative positions(%).

**How to Improve Factuality Score?**  In Equation 3, we propose estimating the factuality of a LLM using Self-Known and Self-Unknown scores. A higher Self-Known score typically corresponds to higher factuality. However, does this mean LLMs would achieve 100% factuality if they had a 100% Self-Known score and 0 Self-Unknown score on their own generation? The answer is no. It is a necessary condition, not a sufficient one for achieving 100% factuality. In the derivation of Equation 3, several additional assumptions are made[6].

According to our results, a higher Self-Known score is usually associated with higher factuality, while a higher Self-Unknown score is associated with lower factuality for LLMs. This indicates that it is challenging for LLMs to recognize unsupported claims on their own. Therefore, a judgment model that incorporates an external knowledge source is necessary for this recognition.

A reasonable question arises: Is the decoding error of LLMs caused by the absence of relevant knowledge? Can Retrieval-Augmented Generation (RAG), which provides additional knowledge, resolve the issue of lower factuality in later stages of generation? In next section, we present some RAG experiments.

### 5.3 RETRIEVAL-AUGMENTED GENERATION

Retrieval-Augmented Generation (RAG) is a widely used approach for enhancing language model performance in various applications. In RAG, relevant text segments are retrieved from an external knowledge source and integrated into the model's responses. For our retrieval corpus, we utilized the English Wikipedia as of April 1, 2023, with each page divided into chunks of up to 256 tokens. These retrieved passages, containing facts relevant to the entity, were incorporated into the LLMs' context to improve the factual accuracy of the generated content.[7].

According to Figure 6a, in the RAG setting, although there are significantly fewer unsupported atomic claims overall, a notable increase in the number of unsupported claims is observed in later stages of generation. As shown in Table 2, LLMs can still response with lots of unsupported claim even given context knowledge. This increase is likely due to error propagation within the LLMs, highlighting the challenges of long-form generation even when relevant parts are provided.

Figure 6a demonstrates that the RAG system exhibits significantly lower Self-Known scores and higher Self-Unknown scores. This discrepancy may stem from the corresponding LLM's lack of prior knowledge regarding the retrieved content in the RAG system, causing it to mistakenly assess accurate information as incorrect.

---

[6]For instance, one key assumption is that the probability of correctness given the model output and input $P(d = 1 \mid \boldsymbol{y}, \boldsymbol{x})$, equals the factuality score $\sigma$ of output $\boldsymbol{y}$, However, if a LLM becomes overconfident in generating answers, the term $P(d = 1 \mid \boldsymbol{y}, \boldsymbol{x})$ may significantly exceed the actual factuality score.

[7]One example is shown in Table 2.

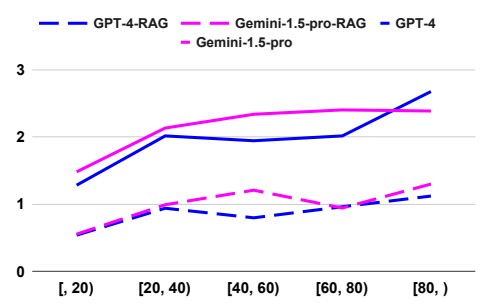 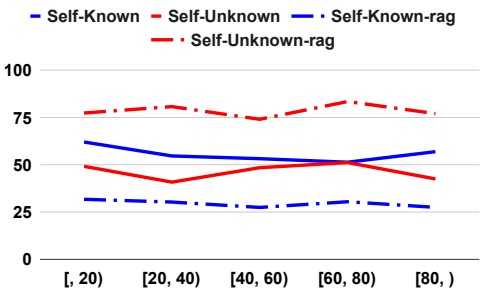

(a) Number of unsupported claims for two LLMs and their RAG models across different relative positions(%).

(b) Self-Known and Self-Unknown scores for Gemini-1.5-pro and the RAG model across different relative positions(%).

In these RAG experiments, when all relevant knowledge is incorporated, LLMs show improvements in factuality. However, they still struggle with lower factuality in later generations. This highlights the need for alternative decoding algorithms specifically designed for long-form generation tasks. Implementing more sophisticated decoding techniques could help mitigate the artifacts associated with long-form generation.

## 6    RELATED WORK

**Factuality Evaluation**    Recent advancements have seen significant efforts in quantifying the factuality of LLM generations. For short answers, factuality often correlates with fact verification, which directly assesses whether the generation aligns with extensive knowledge sources and references (Thorne et al., 2018; Honovich et al., 2022) or utilizes language models (Lin et al., 2022). However, evaluating factuality in long-form content poses greater challenges due to the complexity of the generation process. Recent studies (Fan et al., 2020; Wright et al., 2022; Min et al., 2023) have approached this challenge by breaking down long generations into atomic claims. While these approaches predominantly focus on factual precision, some studies (Wei et al., 2024) also consider evaluating factual recall. In our work, we concentrate on factual precision akin to Min et al. (2023). Moving forward, the development of more robust automatic tools will be crucial for advancing factuality exploration in long-form generation tasks.

**Self-Know and Self-Unknown**    Recent studies have extensively explored the concepts of Self-Known and Self-Unknown in language models. For instance, Kadavath et al. (2022); Liu et al. (2022); Guerreiro et al. (2023) demonstrated that language models are capable of assessing the validity of their own claims and predicting their ability with answering true/false questions accurately. Meanwhile, Srivastava et (2023); Yin et al. (2023) highlighted the limitations of LLMs in acknowledging their unknowns, focusing on their ability to recognize unknown knowledge. In our work, we specifically investigate whether LLMs can identify and reconsider unsupported claims generated from their own outputs. Our results indicate that LLMs struggle to accurately judge unsupported atomic claims from their own generations. We find that a lower Self-Unknown score or a higher Self-Known score corresponds to higher factuality.

## 7    CONCLUSION

In this study, we investigate the factuality of long-form text generation across different model families and relative positions. Our findings reveal a trend of lower factuality in sentences generated later in the sequence. Additionally, we propose methods for enabling LLMs to accurately assess the correctness of atomic claims derived from their own outputs. We introduce an estimation of factuality using Self-Known and Self-Unknown scores, finding that higher Self-Known scores correlate with increased factuality, whereas higher Self-Unknown scores correlate with decreased factuality. These insights provide valuable directions for enhancing factuality in text generation.

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

# A APPENDIX

## A.1 LLMs

Following are the models that are used in our experiments.

- InstructGPT (text-davinci-003) (Ouyang et al., 2022):
- ChatGPT (OpenAI, 2023a)
- PerplexityAI[8]
- GPT-4, GPT-4-Turbo (OpenAI, 2023b)
- Claude-3-Opus (Anthropic, 2024)
- Gemini-1.5-Pro (GeminiTeam, 2024)
- Mixtral-8x7b (Jiang et al., 2024)
- Mistral-Large[9]
- Llama-3-70B-Instruct (AI@Meta, 2024)

## A.2 PROMPT TEMPLATES

### A.2.1 PROMPTS SELF-KNOWN AND SELF-UNKNOWN

**Direct-Asking** Given an atomic claim {claim} and person {person}, we use the following template:

```
Following is a statement from a bio of {person}.
Please check whether the statement is correct or wrong
according to your knowledge.

{claim}
Is this statement true or false?
```

**Question-Answering** Give a question answer pair <{q}, {a}> that is derived from an atomic claim, the following template is used to determine whether LLMs consider the proposed answer is correct:

---

[8] https://www.perplexity.ai/

[9] https://mistral.ai/technology/#models

```
Question: {q}
Proposed Answer: {a}
Is the proposed answer:
   (A) True
   (B) False
The proposed answer is:
```

**Question-Answering w/ None of the above**   Given the question answer pair <{q}, {a}> derived from an atomic claim, the following template is used:

```
Question: {q}
Proposed Answer: {a}
Is the proposed answer:
   (A) True
   (B) False
   (C) None of the above
The proposed answer is:
```

### A.2.2   PROMPTS FOR CREATING THE QUESTION-ANSWER PAIR

Given an atomic claim {claim} of a bio and the person {person}, a question-answer pair can be derived with gpt-4 with the following template:

```
Following is a fact from a bio of {person}.  Please
ask a question and provide the answer.  The answer
is as concise as you can, using a single phrase if
possible.  The answer is also part of the provided
fact.  The question and answer is separetd with #.

{claim}
```

### A.3   RULES FOR FILTERING GENERATIONS

Following are the rules we find that are useful to filter out unresponsive generation.

```
I don't have ...
I do not have ...
I need more information ...
Please provide me ...
Please clarify
I apologize ...
there isn't enough information
Unfortunately, there is no ...
If you can provide more information ...
you could provide more ...
It seems you might ...
```

### A.4   AUTOMATIC TOOL

### A.5   DETAILS ON COMPUTING EXPERIMENTAL RESULT FOR EACH LLM

**Step 1: Obtaining generations**   We feed a prompt "Tell me a bio of <entity>" to the LLM and take the generation. 500 human entities (Min et al., 2023) are used to generate these biographies.

**Step 2: Filtering generations**   For lots of LLMs, a biography is not provided if they think they do not have enough detailed information to provide a biography. We implement rules to filter out these generations[10].

---

[10]The useful rules are shown in Section A.3.

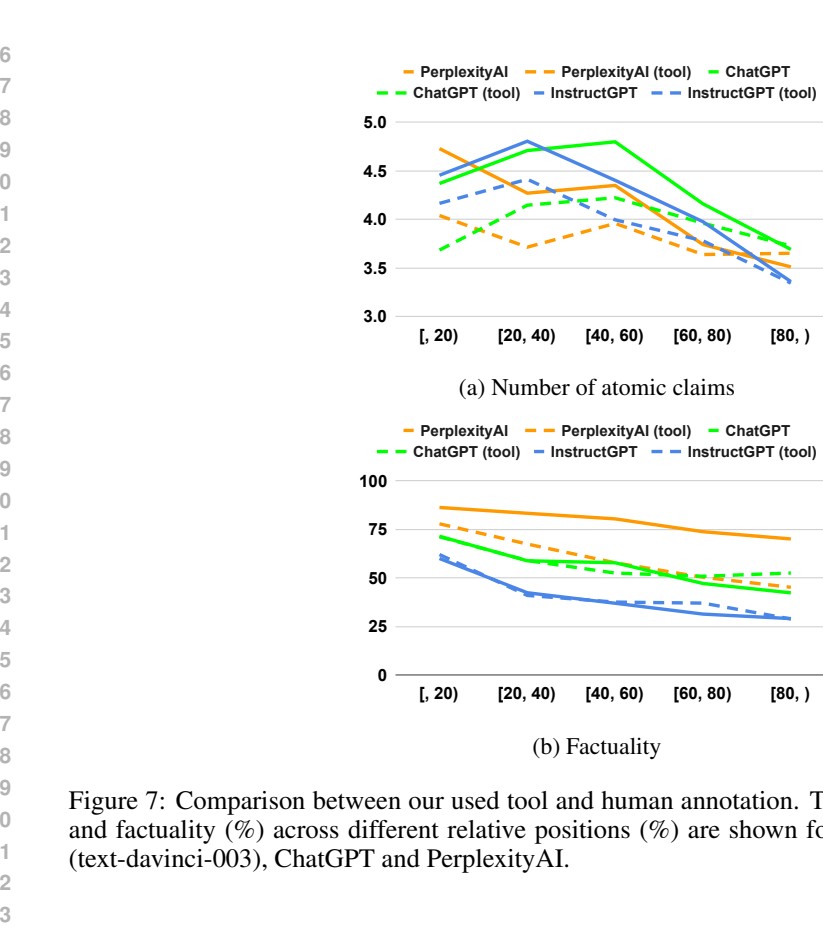

(a) Number of atomic claims

(b) Factuality

Figure 7: Comparison between our used tool and human annotation. The number of atomic claims and factuality (%) across different relative positions (%) are shown for three LLMs: InstructGPT (text-davinci-003), ChatGPT and PerplexityAI.

**Step 3: Evaluation factuality**   We use the tool for breaking generations into atomic claims and evaluate each claim whether it is supported or not. In order to save cost, we randomly sampled 100 samples among the filtered generations. During factuality evaluation, Wikipedia's knowledge source is used in the automatic tool.

**Step 4: Estimation of Self-Known and Self-Unknown**   With above decomposed atomic claims, we use GPT-4 Turbo to get question-answer pairs. For each question-answer pair, a prompt template (see 3 ) is used to determine whether LLMs consider the proposed answer to be correct. The ratios of supported claims judged as correct, and unsupported claims judged as incorrect are then obtained.

## A.6    MORE RESULTS

Table 1: Statistics for various LLMs when generating biographical paragraphs.

|  | #Claims / Gen | Filtered Rate (%) |
| --- | --- | --- |
| GPT-4 | 60.8 | 12.0 |
| Gemini-1.5-pro | 67.5 | 30.0 |
| Claude-3-opus | 41.0 | 42.0 |
| Llama-3-70B-Instruct | 45.9 | 17.2 |
| Mixtral-8x7b | 44.8 | 0.4 |
| Mistral-Large | 48.3 | 5.0 |

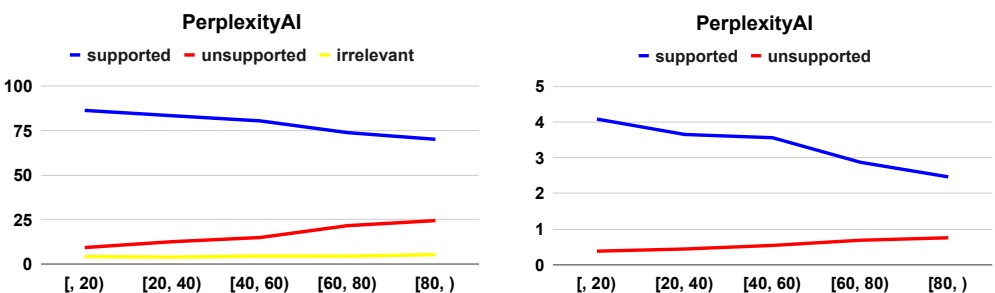

(a) Percentage (%) of supported, unsupported and irrelevant atomic claims.

(b) Number of supported and unsupported atomic claims.

Figure 8: Long-form generation across different relative positions (%) for PerplexityAI.

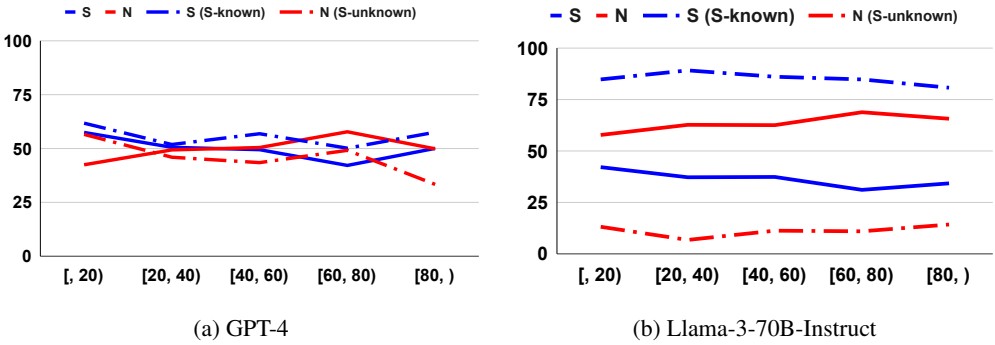

(a) GPT-4

(b) Llama-3-70B-Instruct

Figure 9: Self-Know and Self-Unknown results of different LLMs across different relative positions (%). **S**: **factuality** (percentage of supported atomic claims); **N**: percentage of unsupported atomic claims; **S (S-known)**: percentage of supported atomic claims judged as correct by LLMs; **N (S-unknown)**: percentage of unsupported atomic claims judged as incorrect by LLMs.

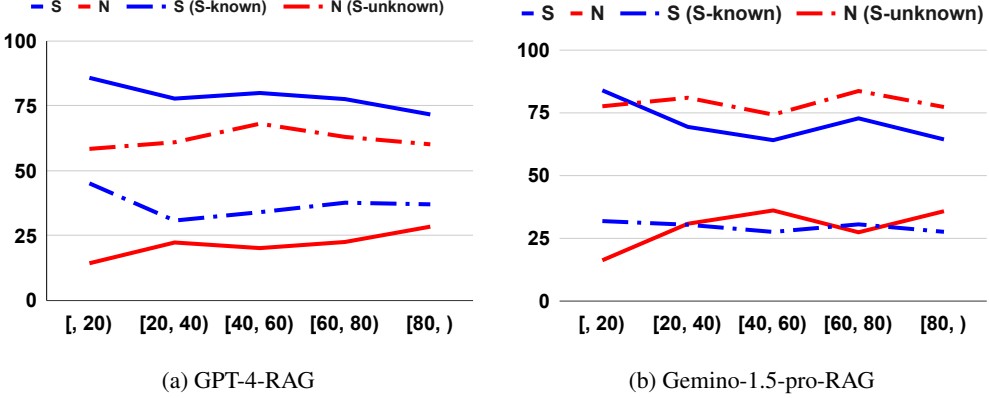

(a) GPT-4-RAG

(b) Gemino-1.5-pro-RAG

Figure 10: Self-Know and Self-Unknown results of different **RAG** models across different relative positions (%). **S**: **factuality** (percentage of supported atomic claims); **N**: percentage of unsupported atomic claims; **S (S-known)**: percentage of supported atomic claims judged as correct by LLMs; **N (S-unknown)**: percentage of unsupported atomic claims judged as incorrect by LLMs.

Table 2: Generated outputs of Gemini-1.5-pro: each document represents a retrieval chunk, with text highlighted in red and yellow indicating unsupported claims. Red text represents claims identified as unsupported by the factuality tool. Yellow text represents unsupported claims that are not identified by the factuality tool.

---

**Document** [0] Jessie Mae Brown Beavers Jessie Mae Brown Beavers (March 18, 1923 – September 6, 1989) was an American journalist based in Los Angeles, California. She was an editor at the "Los Angeles Sentinel" from 1949 to 1989, and served sixteen years on the city's Human Relations Commission, beginning with her 1973 appointment by mayor Tom Bradley.Early life. Jessie Mae Brown was born in Los Angeles, the daughter of Arnetta Hoyt Brown, a Baptist deaconess. She attended the University of California, Los Angeles, where she earned a bachelor's degree in sociology.Career. Brown was editor of the family section of the "California Eagle" from 1944 to 1949, when she joined the staff of the "Los Angeles Sentinel" as an editor. In 1966 she was one of the organizers and leaders of the Los Angeles chapter of the National Association of Media Women. In 1969, she was given the Outstanding Woman in Journalism Award by the University of Southern California chapter of Theta Sigma Phi. Beavers served on the Los Angeles Human Relations Commission for sixteen years, after she was appointed by mayor Tom Bradley in 1973. On the commission, she worked closely with fellow commissioner Toshiko S. Yoshida during the American bicentennial

**Document** [1] , chaired the affirmative action subcommittee, and in 1982 organized hearings on racial bias in the entertainment industry. She also served one the Los Angeles County Music and Performing Arts Commission. She ran for a seat on the Los Angeles City Council in 1987. Beavers was a member of Jack and Jill and the Lullaby Guild. In 1968 she was cited for her work by the President's Youth Council. In 1972 she was honored alongside Quincy Jones, editor Ruth Washington, and student leader Willis Edwards by the Committee of Women for Good Government. In 1973 she was honored by colleagues and admirers (including Mayor Bradley, judge Vaino Spencer, columnist Gertrude Gipson, and clubwoman Leontyne Butler King) at a tribute event at the Beverly Wilshire Hotel, hosted by singer Thelma Houston.Personal life. In 1947 Brown married insurance executive Leroy A. Beavers, Jr., nephew of Los Angeles businessman George A. Beavers Jr. Actress Dorothy Dandridge was one of her bridesmaids in the wedding. They had three children. In 1979, she was assaulted and robbed in the driveway of her home. She died in 1989, aged 66 years, in Los Angeles. Her grave is at Forest Lawn Cemetery in Gl

**Document** [2] endale, California.

---

**Question:** Tell me a bio of Jessie Mae Brown Beavers.

---

Response: Jessie Mae Brown Beavers (March 18, 1923 – September 6, 1989) was a prominent American journalist and community leader based in Los Angeles, California. Born in Los Angeles, she obtained her bachelor's degree in sociology from the University of California, Los Angeles.

Beavers's career was deeply intertwined with journalism and activism. She served as an editor for the *California Eagle* and later the *Los Angeles Sentinel*, a prominent African American newspaper, for over four decades. Her commitment to social justice led her to co-found the Los Angeles chapter of the National Association of Media Women and actively advocate for fair representation in the media.

Appointed by Mayor Tom Bradley, Beavers dedicated sixteen years to the Los Angeles Human Relations Commission, tackling issues of racial bias and promoting affirmative action. Her influence extended to her service on the Los Angeles County Music and Performing Arts Commission.

Throughout her life, Beavers received numerous accolades, including recognition from the President's Youth Council and a tribute event graced by notable figures like Mayor Bradley and singer Thelma Houston. Her enduring legacy continues to inspire aspiring journalists and advocates alike.

---

