# OpenReview forum: "Investigating Factuality in Long-Form Text Generation: The Roles of Self-Known and Self-Unknown"
_ICLR.cc/2025/Conference — Submitted to ICLR 2025_

### Official Review · Reviewer_RXNh · 2024-10-18

**Soundness:** 2
**Presentation:** 2
**Contribution:** 1
**Rating:** 3
**Confidence:** 3

**Summary:**

This paper investigate the factuality in long-form text generation. Using the existing datasets, it defines the Self-Known and Self-Unknown scores judged by the corresponding LLMs themselves. Based on several sets of statistics and empirical experiments, the authors draw several findings which might be useful for the community.

**Strengths:**

1. Long-form factuality is a promising research direction.

**Weaknesses:**

1. The statistical observations are based on an existing dataset of Biography from Min et al., 2023, which impair the contribution of this work.

2. The key finding of this work, *factuality scores tend to decline in later sentences of the generated text, accompanied by a rise in the number of unsupported claims*, is obvious and not exciting. This is a general challenge for LLMs to generate long-form texts in the later parts in terms of factuality, coherency, creativity, etc.

3. This work does not provide any insightful reasons, either. As stated in Section 2.1, *We hypotheses the possible reasons are the error propagation and these generated claims are with low confidence by LLMs*, this is not convincing.

4. The results presented in the main text overlap somewhat. For example, I don't quite get why reporting both the percentage and number of supported and unsupported atomic claims in Figure 1.

5. It is abrupt to discuss retrieval-augmented generation in Section 5.3. I don't quite get the necessity of introducing RAG.

6. The presentation is not good, which should be proofread one more time. For example, the duplicate descriptions of *Mixtral* in line 18 and *in this work* in line 42. Line 59: *Our findings indicate that sentences generated **later** in the sequence generally demonstrate higher factuality.* I think it should be *sentences generated **early** in the sequence generally demonstrate higher factuality*?

**Questions:**

See above.

---

> ### Author Response · Authors · 2024-11-20
>
> Thank you for the review and helpful comments!
>
> ### **The statistical observations are based on an existing dataset of biography from Min et al., 2023, which impairs the contribution of this work.**
>
> We appreciate the reviewer’s concern regarding the use of an existing dataset, and we understand the importance of distinguishing our contributions. While we did use the dataset from Min et al. (2023) for our experiments, we believe our work still makes a significant contribution in the following ways:
>
> - Our primary contribution lies not in the dataset itself but in the novel findings to the factuality evaluation of long-form text generation. By analyzing self-known and self-unknown metrics and their correlation with factuality errors, we provide new insights into how LLMs handle factuality across different segments of long-form text, which has not been explored in this manner before.
>
> - The proposed mathematics linking from factuality to self-known and self-unknown scores, and the related empirical results — offers a quantifiable method to assess the accuracy of generated content. This approach provides a fresh perspective on evaluating LLM outputs and sets the stage for future improvements in error detection and correction.
>
> - We agree with the reviewer that validating the findings on additional datasets strengthens the robustness of the conclusions. However, there is a limited availability of human-annotated, long-form LLM factuality datasets. We aim to apply this to more diverse datasets in future work to strengthen the generalizability of our findings.
>
>
>
>
> ### **Factuality scores tend to decline in later sentences is obvious and not exciting.**
>
> We appreciate the reviewer’s observation regarding the tendency of models to exhibit more factuality errors in the later parts of their outputs. Our findings confirm that current state-of-the-art LLMs, such as GPT-4 and Claude-3-Opus, still face this issue.
> Our work aims to advance this understanding by providing a quantitative analysis that correlates these errors with Self-Known and Self-Unknown scores, as detailed in Equation (3). This analysis introduces a novel perspective by mathematically linking these metrics to factuality, offering deeper insights into the underlying causes of these errors and potential mitigation strategies.
> Additionally, we observe that for advanced LLMs, the Self-Known scores show minimal variation across positions. However, factuality remains lower in later sections, suggesting a strong positional bias in generation. In the RAG experiments, even when all relevant knowledge is incorporated, LLMs still struggle with lower factuality in later generations. This highlights the need for alternative decoding algorithms specifically designed for long-form generation tasks.
>
> ### **This work does not provide any insightful reasons**
>
> We developed a mathematical framework linking Self-Known and Self-Unknown scores to factu- ality, providing deeper insights into their relationship. Both empirical and theoretical results demon- strate that higher Self-Known scores correspond to improved factuality, while higher Self-Unknown scores are associated with reduced factuality. Moreover, even without significant changes in the models’ self-judgment (Self-Known and Self-Unknown), the number of unsupported claims may still increase, likely reflecting the inherent challenges of long-form generation.
>
>
> ### **Why reporting both the percentage and number of supported and unsupported atomic claims in Figure 1**
>
> Figure 1(a) illustrates the percentage of supported and unsupported claims across different relative positions, while Figure 1(b) presents the absolute number of supported and unsupported claims at these positions. Both views are provided to account for variations in the total number of claims across different relative positions.
>
> ### **Necessity of introducing RAG.**
>
> In the RAG experiments, even when all relevant knowledge is incorporated, LLMs still struggle with lower factuality in later generations. This highlights the need for alternative decoding algorithms specifically designed for long-form generation tasks.
>
>
> ### **Correction of Errors and proof reading**
>
> We have corrected the phrasing in lines 60 and 85 to ensure consistency with our actual findings. Specifically, we state that "sentences generated later in the sequence tend to exhibit lower factuality scores.
>
> We have conducted a thorough proofreading of the manuscript to identify and resolve similar inconsistencies, ensuring accuracy throughout.

---

> > ### Comment · Reviewer_RXNh · 2024-11-21
> >
> > Thanks the authors for the response. I have the following comments:
> >
> > 1. Since the partial contribution of this work is to provide the statistical observations, authors should ensure their conclusions are robust. If there are exiting datasets available for statistical observations, that would be great to test on as many datasets as you can. If not, authors should address this issue, for example, create a dataset for the community. This would enhance the contribution of this work. Although there is a limited availability of human-annotated, long-form LLM factuality datasets, testing on a single one cannot provide significant findings.
> >
> > 2. Based on the above reason, it is unclear whether the analysis by mathematically linking these metrics to factuality still holds to understand the underlying causes.
> >
> > 3. I still cannot understand the motivation to discuss on RAG. If it is indeed necessary, the authors should motivate it as early as possible in the introduction.
> >
> > 4. The text explaining the insightful reason in the response appears to be copied from the paper, contains punctuation errors, and does not further explain the contribution of the paper. Authors should proofread the response during rebuttal and explain more details for helping reviewers understand.

---

> > > ### Author Response · Authors · 2024-11-22
> > >
> > > Thanks for your quick response!
> > >
> > > 1. We acknowledge the limitation of using a single dataset. However, this dataset is sufficient to study the problem across different advanced LLM families. It effectively highlights the limitations and serious issues inherent in current long-form text generation. To further strengthen our contribution, we developed a mathematical framework that links Self-Known and Self-Unknown scores to factuality.
> > >
> > > 2. The mathematical framework linking these metrics to factuality is sufficient to establish a correlation. The empirical results presented in Figure 3, based on a single dataset, are used to validate this mathematical relationship.
> > >
> > > 3. Motivation for RAG (Retrieval-Augmented Generation) is to isolate the source decoding error from the absence of relevant knowledge. The RAG experiments demonstrate that supplying additional relevant knowledge can help improve the factuality. Yet, it fails to fully address the issue of poorer factuality observed at later positions of the generated sequence. This highlights the need for alternative decoding algorithms specifically designed for long-form generation tasks. A paragraph discussing the motivation for RAG has been added in Section 5.2 on page 8.
> > >
> > > 4. The insightful reason was updated previously. Thank you for pointing out the punctuation errors—we have noted them and made corrections. We are continuing to work on additional proofreading.

---

### Official Review · Reviewer_8SW2 · 2024-11-02

**Soundness:** 2
**Presentation:** 1
**Contribution:** 2
**Rating:** 3
**Confidence:** 3

**Summary:**

The paper explores the challenges large language models (LLMs) face in maintaining factual accuracy in extended text outputs, analyzing models like GPT-4, Gemini-1.5-Pro, and Llama-3-70B. The author first estimate the factuality score based on ‘’Self-Known and Self-Unknown’, and the evaluation shows that the score is well aligned with human evaluation. The study shows that factuality often declines over longer text, with later sentences containing more unsupported claims. Using new metrics, Self-Known, Self-Unknown and factuality score, the authors assess how well LLMs recognize their own factuality; they find that even advanced models have limited ability to self-assess accurately, achieving only modest success in identifying correct or incorrect claims. The findings emphasize the need for improved self-assessment in LLMs to enhance factual consistency, particularly in long-form generation.

**Strengths:**

a.	The problem explored in the paper is interesting, and worth further exploration.
b.	The authors estimate a factuality score based on Self-Known and Self-Unknown, which makes sense

**Weaknesses:**

a.	The presentation of the paper is confusing, the logic line of the paper is very unclear. For example, the paper proposes to use three ways to evaluate the self-known and self-unknown, however, no details is provided for all the methods. And in Section 4, an estimation of the factuality score is proposed, and in section 5, they mentioned that they measure the factuality score using FActScore, then which score do they used in the following experiments?
b.	As an analytical paper, it lacks insightful findings or conclusions. It has been found in some previous works that the model generates more unsupported atomics in the later sentences, and this is a known problem with long-form generation.
c.	 Some errors in the paper writing,  e.g. in line 60, they said ‘Our findings indicate that sentences generated later in the sequence generally demonstrate higher factuality.’ and in line 85, they said ‘...LLMs typically exhibit lower factuality scores in the later segments of long-form text.’ , which are contradictory.

**Questions:**

a.	Which factuality score did you use for evaluation, the FActScore or the estimation in section 4?
b.	Can you elaborate more on the assumption in line 250?

---

> ### Author Response · Authors · 2024-11-20
>
> We appreciate the reviewer’s feedback on the clarity of the paper’s presentation and the logical flow of the methodology.
>
> ### **Clarifying the Logical Flow**
>
> In Section 3, we explore three ways to evaluate the Self-known and Self-Unknown. The beginning of Page 4 shows the three approaches, and three corresponding prompt templates provided in Appendix Section A.2, which is already shown.
>
> In Section 4, we provide a novel perspective by mathematically linking Self-known and Self-Unknown scores  to factuality, offering deeper insights into the underlying causes of these errors and potential mitigation strategies.
>
> In Section 5,  we provide the empirical results on different LLMs on Self-Known, Self-Unknown scores  and factuality.
>
> ### **Clarifying Factuality Score Usage**
>
> We have clarified in Section 4 that the estimation of the factuality score provides a theoretical foundation, while in Section 5.1, automatic tool is used for practical implementation. Additionally, at the beginning of Section 5, we have explicitly stated which factuality score is used in subsequent experiments, ensuring consistency and transparency throughout the paper.
>
> ### **Novel Insights and conclusion**
>
> We appreciate the reviewer’s observation regarding the tendency of models to exhibit more factuality errors in the later parts of their outputs. Our findings confirm that current state-of-the-art LLMs, such as GPT-4 and Claude-3-Opus, still face this issue.
> Our work aims to advance this understanding by providing a quantitative analysis that correlates these errors with Self-Known and Self-Unknown scores, as detailed in Equation (3). This analysis introduces a novel perspective by mathematically linking these metrics to factuality, offering deeper insights into the underlying causes of these errors and potential mitigation strategies.
> Additionally, we observe that for advanced LLMs, the Self-Known scores show minimal variation across positions. However, factuality remains lower in later sections, suggesting a strong positional bias in generation. In the RAG experiments, even when all relevant knowledge is incorporated, LLMs still struggle with lower factuality in later generations. This highlights the need for alternative decoding algorithms specifically designed for long-form generation tasks.
>
>
> ### **Correction of Errors and proof reading**
>
> We have corrected the phrasing in lines 60 and 85 to ensure consistency with our actual findings. Specifically, we state that "sentences generated later in the sequence tend to exhibit lower factuality scores.
>
> We have conducted a thorough proofreading of the manuscript to identify and resolve similar inconsistencies, ensuring accuracy throughout.
>
>
> ### **Assumption on log-likelihood of generation (true and false)**
> We thank the reviewer for raising the issue regarding the assumption on log-likelihood in our model. To clarify, our framework makes the following assumptions about the log-likelihoods of both true and false generation outputs: Since the true and false parts of the whole generated output come from the same model, their log-likelihoods should be similar, as shown below:
>
>
> \begin{equation}
>     P(y \mid x) \approx P(\mathrm{y_{correct}} \mid x) \approx P(\mathrm{y_{wrong}} \mid x) \nonumber
> \end{equation}

---

### Official Review · Reviewer_ksoE · 2024-11-03

**Soundness:** 3
**Presentation:** 2
**Contribution:** 3
**Rating:** 5
**Confidence:** 3

**Summary:**

This paper explores the issue of factuality in long-form text generation by large language models (LLMs), specifically addressing their tendencies to produce a mix of true and false information. The study examines six advanced LLMs, including GPT-4, Claude-3-Opus, and Mixtral, to investigate how factuality scores evolve throughout the text. The authors propose two novel evaluation metrics, Self-Known and Self-Unknown, to measure whether LLMs can accurately self-assess the correctness of their generated content. Results reveal that while LLMs generally perform well in earlier parts of the text, factuality tends to decline in later sections. Additionally, higher Self-Known scores correlate with better factuality, while higher Self-Unknown scores indicate the opposite. The authors also test various evaluation settings, emphasizing the importance of incorporating external knowledge to improve models’ factual accuracy.

**Strengths:**

The introduction of Self-Known and Self-Unknown scores provides a fresh perspective on evaluating LLMs' self-assessment capabilities, offering valuable insights into their strengths and limitations. The paper systematically analyzes multiple LLMs, covering different evaluation settings and utilizing both human and automated tools for factuality assessment. This extensive examination adds rigor to the study. By highlighting the consistent decline in factuality in long-form outputs, the research addresses a critical challenge for deploying LLMs in real-world applications, making the findings highly relevant for model improvement. The observed relationships between factuality and self-assessment metrics could guide future work on enhancing LLMs’ self-awareness and knowledge calibration, which is vital for developing reliable generative AI models.

**Weaknesses:**

1. The difference between Figure 1 (a) and Figure 1 (b) is not illustrated. It will be beneficial to add illustrations in caption.
2. The proposed method can perform evaluation to long-term generation by the model itself, but it is time-consuming to do evaluation for every atomic statement. The detection itself has a very strong trade-off between performance and inference costs.
3. There is not a way to adjust or convert the fact errors into correct facts. The proposed method only covers fact error detection.
4. Missing results for LLAMA family makes the results less comprehensive.
5. The scope of the proposed method is a bit limited as it can only be applied to factuality tasks which can be separated into atomic statements. However, some factuality tasks may not explicitly contains atomic statements.

**Questions:**

As in weakness.

---

> ### Author Response · Authors · 2024-11-20
>
> Thank you for the review and helpful comments!
>
> ### **The difference between Figure 1 (a) and Figure 1 (b) is not illustrated.**
>
> Figure 1(a) illustrates the **percentage of supported and unsupported claims** across different relative positions, while Figure 1(b) presents the **absolute number of supported and unsupported claims** at these positions. Both views are provided to account for variations in the total number of claims across different relative positions. Please see the caption in the Figure 1.
>
>
> ### **“Time-consuming to do evaluation for every atomic statement”** and  **“a bit limited as it can only be applied to factuality tasks which can be separated into atomic statements.”**
>
> Yes, evaluating every atomic statement can be time-consuming, but it provides a fine-grained assessment, especially since LLM responses often contain a mix of both true and false information. Currently, this level of atomic evaluation is the standard approach for assessing long-form factuality, as demonstrated in works such as [1,2].
>
> [1] FACTSCORE: Fine-grained Atomic Evaluation of Factual Precision in Long Form Text Generation. EMNLP 2023
>
> [2] Long-form factuality in large language models. Neurips 2024
>
>
> ### **“There is not a way to adjust or convert the fact errors into correct facts. The proposed method only covers fact error detection.”**
>
> We thank the reviewer for highlighting this important point. Indeed, the primary focus of our work is on detecting and analyzing factual errors rather than directly correcting them. The goal of the proposed method is to provide a robust framework for identifying and quantifying factual inaccuracies in long-form generation, which we believe is a critical step toward improving factual consistency in LLM outputs.
>
> ### **“Missing results for the LLAMA family makes the results less comprehensive.”**
>
>  Please see Figure 9 in the appendix. More related results are already shown.

---

### Official Review · Reviewer_3uij · 2024-11-04

**Soundness:** 3
**Presentation:** 3
**Contribution:** 3
**Rating:** 5
**Confidence:** 3

**Summary:**

This paper studied factuality in long text generation. It studies the position of the atomic fact and finds that the latter part of the text generated is more likely to be not factual. It invented two metrics: self-Known (the percentage of correct atomic facts by predicted as correct atomic facts) and Self-Unknown (the percentage of incorrect atomic facts are correctly predicted as incorrect). The authors claim the two metrics can help measure the true factuality score.

**Strengths:**

- The authors did extensive analysis to support our intuition that there tends to more factual errors in the latter part of the LLM's response. The authors show this applies to different models and also applies to RAG scenario.

**Weaknesses:**

- The Finding where model tends to have more factuality errors in later part of the output is kind of accepted conclusion. So the findings is no surprising
- The conclusion about the relationship between factuality score and Self-Known and Self-Unknown is not established. Firstly, I think there is something wrong when transiting from Eq (1) to Eq (2) and it is hard to understand the definition of "y_correct" and "y_wrong". Secondly, the assumption that there is no "over confidence" is not correct.  Lastly, it will be helpful if authors can verified the conclusion in Figure 3 in more than 1 datasets.

**Questions:**

1. what are definitions of  the definition of "y_correct" and "y_wrong? can you show why Eq (1) can transit to Eq (2)?
2. I think "self-Known" is also called, True Negative Rate (if we consider the labe of being not-factual as positive) and "Self-Unknownw" will be "True Postiive Rate". Is this correct?

---

> ### Author Response · Authors · 2024-11-20
>
> Thank you for the review and helpful comments!
>
> ### **"The Finding where model tends to have more factuality errors in later part is no surprising"**
>
> We appreciate the reviewer’s observation regarding the tendency of models to exhibit more factuality errors in the later parts of their outputs as a well-known phenomenon. Our findings confirm that current state-of-the-art LLMs, such as GPT-4 and Claude-3-Opus, still face this issue.
>
> Our work aims to advance this understanding by providing a quantitative analysis that correlates these errors with Self-Known and Self-Unknown scores, as detailed in Equation (3). This analysis introduces a novel perspective by mathematically linking these metrics to factuality, offering deeper insights into the underlying causes of these errors and potential mitigation strategies.
>
> Additionally, we observe that for advanced LLMs, the Self-Known scores show minimal variation across positions. However, factuality remains lower in later sections, suggesting a strong positional bias in generation. In our RAG experiments, even when all relevant knowledge is included, LLMs still struggle with factuality issues. These observations highlight the need for alternative decoding algorithms tailored specifically for long-form generation tasks.
>
>
>
>
> ###  **Steps from Eq (1) to Eq (2)**
>
> Thanks for point about the confusion. Following is the rewritten steps:
> $$
> \begin{aligned}
> & \sigma * P(y \mid x)  \\\\
> =& P(d=1 \mid y, x)* P(y \mid x) = P(d=1,y \mid
> x)  \nonumber \\\\
> =& P(d=1, \mathrm{y_{correct}} \mid x) \sigma + P(d=1, \mathrm{y_{wrong}} \mid x) (1-\sigma) \nonumber \\\\
> =& P(d=1 \mid \mathrm{y_{correct}} ) P(\mathrm{y_{correct}} \mid x) \sigma + P(d=1 \mid \mathrm{y_{wrong}} ) P(\mathrm{y_{wrong}} \mid x) (1-\sigma)
> \end{aligned}
> $$
>
> These updates can be found in Section 4 on page 5 of the revised manuscript. We hope these changes address the reviewer’s concerns and make the derivation more comprehensible.
>
> ### **Definition of "y_correct" and "y_wrong"**
>
> We acknowledge that the definitions of 'y_correct' and 'y_wrong' were not sufficiently clear in the original manuscript. To address this, we have revised the definitions to explicitly state that 'y_correct' refers to model outputs aligned with the ground truth and 'y_wrong' refers to outputs that are wrong.
>
>
> ### **Assumption of No 'Overconfidence**
>
> We appreciate the reviewer’s comment regarding the assumption of no "overconfidence." While this assumption was introduced to simplify the mathematical model, we agree that it does not fully capture the complexities of real-world scenarios. To address this:
> We have clarified the scope and limitations of this assumption in Section related work.
> Some existing work[1,2,3] already demonstrated that LLMs are capable of assessing the validity of their own claims and predicting their ability with answering true/false questions accurately.
>
> [1] Language models (mostly) know what they know. Anthropic. 2022.
>
> [2] A token-level reference-free hallucination detection benchmark for free-form text generation. ACL 2022
>
> [3] Looking for a needle in a haystack: A comprehensive study of hallucinations in neural machine translation. EACL 2023
>
>
> ### **Validation on More Than One Dataset**
>
> We agree with the reviewer that validating the findings on additional datasets strengthens the robustness of the conclusions. However, there is a limited availability of human-annotated, long-form LLM factuality datasets.
>
> To address this limitation: we have included a footnote in Section 4 on page 6 about the current scarcity of such datasets and include details for research reproducibility and validation. Future work will aim to address this gap as more annotated datasets become available.
>
>
> ### **Self-Known is also called, True Negative Rate and Self-Unknown" will be True Positive Rate**
>
> Yes, they look similar. This is another way to interpret these two scores.

---

### Meta-Review · Area_Chair_68YV · 2024-12-25

**Metareview:**

This paper investigates the factuality of long-form text generation by large language models (LLMs), revealing that factuality declines in later sections of generated text. Introducing two metrics, Self-Known and Self-Unknown, the study assesses LLMs' ability to self-evaluate the correctness of their outputs, finding that even advanced models like GPT-4 exhibit limitations in self-assessment. The findings highlight the need for improved self-evaluation mechanisms to enhance factual consistency in long-form generation.

Strength:
Studying the factuality problem in LLM is an important topic, especially how it evolves over the long-form generation situations.

Weakness:
The main weakness of this paper is that it provides relatively less insights about the problem. The factuality decays over at the later positions of long-form generation is not surprising. And defining the two metrics measuring their correlations are not a significant enough contribution. In addition, the work is only carried out on one dataset and should be evaluated on more datasets. As reviewers pointed out that testing the claims on one dataset cannot provide significant findings.

Overall, due to limited contribution and insights provided by this work, this paper cannot be accepted in its current form.

**Additional Comments On Reviewer Discussion:**

The authors has addressed some of the clarification problems raised by the reviewers. However, the response regarding the main concern (not providing insightful reasons for the observation on decaying factuality over the generation process) has not been well addressed in the rebuttal.

---

### Decision · Program_Chairs · 2025-01-22

Reject